# A shared ancient enhancer element differentially regulates the *bric-a-brac* tandem gene duplicates in the developing *Drosophila* leg

Henri-Marc G. Bourbon[1☯], Mikhail H. Benetah[1☯], Emmanuelle Guillou[1], Luis Humberto Mojica-Vazquez[1¤a], Aissette Baanannou[1¤b], Sandra Bernat-Fabre[1], Vincent Loubiere[2¤c], Frédéric Bantignies[2], Giacomo Cavalli[2], Muriel Boube[1¤d]*

**1** Center for Integrative Biology, Molecular Cellular and Developmental (MCD) Biology Unit, Federal University of Toulouse, Toulouse, France, **2** Institute of Human Genetics, University of Montpellier, CNRS Montpellier, France

☯ These authors contributed equally to this work.
¤a Current address: Genotoxicología Ambiental, Departamento de Ciencias Ambientales, Centro de Ciencias de la Atmósfera, Universidad Nacional Autónoma de México, México, México
¤b Current address: Laboratory of Molecular and Cellular Screening Processes, Center of Biotechnology of Sfax, Sfax, Tunisia
¤c Current address: Institute of Molecular Pathology (IMP), Vienna BioCenter (VBC), Vienna, Austria
¤d Current address: RESTORE Research Center, Université de Toulouse, INSERM 1301, CNRS 5070, EFS, ENVT, Toulouse, France
* muriel.boube-trey@inserm.fr

**Data Availability Statement:** All relevant data are within the manuscript and its Supporting Information files.

## Abstract

Gene duplications and transcriptional enhancer emergence/modifications are thought having greatly contributed to phenotypic innovations during animal evolution. Nevertheless, little is known about how enhancers evolve after gene duplication and how regulatory information is rewired between duplicated genes. The *Drosophila melanogaster bric-a-brac* (*bab*) complex, comprising the tandem paralogous genes *bab1* and *bab2*, provides a paradigm to address these issues. We previously characterized an intergenic enhancer (named LAE) regulating *bab2* expression in the developing legs. We show here that *bab2* regulators binding directly the LAE also govern *bab1* expression in tarsal cells. LAE excision by CRISPR/Cas9-mediated genome editing reveals that this enhancer appears involved but not strictly required for *bab1* and *bab2* co-expression in leg tissues. Instead, the LAE enhancer is critical for paralog-specific *bab2* expression along the proximo-distal leg axis. Chromatin features and phenotypic rescue experiments indicate that LAE functions partly redundantly with leg-specific regulatory information overlapping the *bab1* transcription unit. Phylogenomics analyses indicate that (i) the *bab* complex originates from duplication of an ancestral singleton gene early on within the Cyclorrhapha dipteran sublineage, and (ii) LAE sequences have been evolutionarily-fixed early on within the Brachycera suborder thus predating the gene duplication event. This work provides new insights on enhancers, particularly about their emergence, maintenance and functional diversification during evolution.

**Funding:** Research in the HMB-MB laboratory was supported by grants from the Association pour la Recherche sur le Cancer (ARC PJA 20141201932) to MB, the Agence Nationale de la Recherche (ANR-16 CE12-0021-01) to MB, and institutional basic support from of the Centre National de Recherche Scientifique (CNRS) to HMB and Toulouse III University to HMB. Research in the G. C. laboratory was supported by a grant from the European Research Council (Advanced Grant 3DEpi) and by the CNRS (for GC and FB). MHB obtained a PhD fellowship from the French « Ministère de L'Enseignement Supérieur et de la Recherche », LHMV from the Mexican CONACYT and AB from a CNRS- Conseil Régional Midi-Pyrénées co-financing and then from the ARC. V.L. was supported by a doctoral fellowship from the Laboratory of Excellence EpiGenMed and the ARC. The funders had no role in study design, data collection and analysis, decision to publish, or preparation of the manuscript.

**Competing interests:** The authors have declared that no competing interests exist.

## Author summary

During animal evolution, de novo emergence and rewiring of transcriptional enhancers have contributed to morphological innovations. However, how enhancers regulate distinctly gene duplicates and are evolutionary-fixed remain largely unknown. The *Drosophila bric-a-brac* (*bab*) locus, comprising the tandemly-duplicated genes *bab1* and *bab2*, provides a good paradigm to address these issues. In this study, genetic analyses show a partial co-regulation of both genes in the developing leg depending on tissue-specific transcription factors known to bind an intergenic enhancer. Genome editing reveals that this enhancer is shared by both genes and is also critically required for *bab2*-specific expression. Chromatin features and phenotypic rescue experiments indicate the existence of partly-redundant limb-specific regulatory information within the *bab1* transcription unit. Phylogenomics analyses among Diptera indicate that the *Drosophila bab* locus originates from duplication of a singleton gene within the Brachycera lineage. Lastly, we show that whereas *bab1* promoter and leg enhancer sequences have been well conserved after the duplication event, *bab2* promoter and other *bab* enhancers have evolved more recently in the Cyclorrhapha sublineage. This work brings some new insights about (i) how a single enhancer can drive specificity among tandem gene duplicates, and (ii) how enhancers evolutionary adapt with distinct cognate gene promoters.

## Introduction

Gene duplications have largely contributed to create genetic novelties during evolution [1,2]. Intra-species gene duplicates are referred to as "paralogs", which eventually diverged functionally during evolution in a phylogenetic manner. Gene family expansion has facilitated phenotypic innovation through (i) acquisition of new molecular functions or (ii) the subdivision of the parental gene function between the duplicate copies [3–5]. Phenotypic novelties are thought having originated from both modifications of protein sequences and evolutionary emergence or modifications of genomic <u>C</u>is-<u>R</u>egulatory <u>E</u>lements (CREs) or modules, most often dubbed as "enhancer" regions, which regulate gene transcription in a stage-, tissue- and/or cell-type-specific manner [6–10]. While many shared CRE/enhancers have been described in *Drosophila* for several gene complexes [11–14], how they emerge and are differentially evolving remain largely elusive.

The ~150-kilobase (kb) long *Drosophila melanogaster bric-a-brac* (*bab*) locus, located on the third chromosome (3L arm), comprises two tandemly-duplicated genes (Fig 1A), *bab1* and *bab2*, which encode paralogous transcription factors sharing two conserved domains: (i) a <u>B</u>ric-a-brac/<u>T</u>ramtrack/<u>B</u>road-complex (BTB) domain involved in protein-protein interactions, and (ii) a specific DNA-binding domain (referred to as BabCD, for <u>Bab</u> <u>C</u>onserved <u>D</u>omain), in their amino(N)- and carboxyl(C)-terminal moieties, respectively [15]. Bab1-2 proteins are co-expressed in many tissues [15,16]. In the larval epidermis, they co-regulate directly *yellow* expression in a sexually-dimorphic manner, thus controlling adult male versus female body pigmentation traits [17–20]. *bab1-2* co-expression in the developing epidermis is partially governed by two CREs which drive reporter gene expression (i) in a monomorphic pattern in the abdominal segments A2-A5 of both sexes (termed AE, for "<u>A</u>nterior <u>E</u>lement"), and (ii) in a female-specific pattern in the A5-A7 segments (DE, for "<u>D</u>imorphic <u>E</u>lement") (Fig 1A) [18,21]. In addition to controlling male-specific abdominal pigmentation traits, *bab1-2* are required, singly, jointly or in a partially-redundant manner, for embryonic cardiac

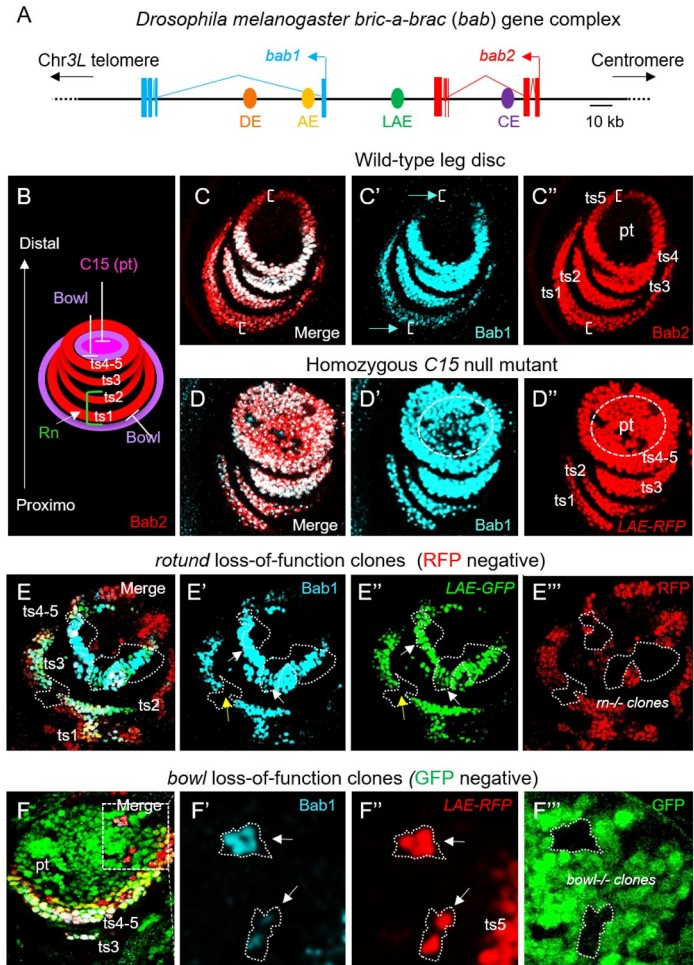

**Fig 1. *C15*, *rotund* and *bowl* regulate both *bab1* and *bab2* expression. (A)** Schematic view of the *Dmel bab* locus on the 3L chromosomal arm (Chr*3L*). The tandem *bab1* (blue) and *bab2* (red) transcription units (filled boxes and broken lines represent exons and introns, respectively), the previously known CRE/enhancers are depicted by filled dots (abdominal DE and AE in dark and light orange, respectively; leg/antennal LAE in dark green and cardiac CE in purple), and the telomere and centromere directions are indicated by arrows. **(B)** Scheme depicting C15, Bowl and Rn TF activities in regulating *bab2* expression as a four-ring pattern within the developing distal leg. **(C)** Medial confocal view of a wild-type L3 leg disc. Merged Bab1 (cyan) and Bab2 (red) immunostainings, as well as each marker in isolation in (C') and (C"), respectively, are shown. Positions of *bab2*-expressing ts1-5 cells and the pretarsal (pt) field are indicated in (C"). Brackets indicate paralog-specific *bab2* expression in ts1 and ts5 cells, and blue arrows corresponding cell rows that do not express *bab1*. **(D)** Distal confocal view of a homozygous *C15²* mutant L3 leg disc expressing *LAE-RFP^{ZH2A}*. Merged Bab1 immunostaining (in cyan) and RFP fluorescence (red), and each marker in isolation in (D') and (D"), are shown. Bab2-expressing mutant pt cells are circled with a dashed line in (D') and (D"). **(E)** Medial confocal view of a mosaic L3 leg disc expressing *LAE-GFP^{ZH2A}* and harboring *rotund* mutant clones. Merged Bab1 (cyan) immunostaining, GFP (green) and RFP (red) fluorescence, as well as each marker in isolation in (E'), (E") and (E'''), respectively, are shown. Mutant clones are detected as black areas, owing to the loss of RFP. The respective ts1-5 fields are indicated in (E). White arrows indicate *rotund-/-* clones still expressing *bab1* and yellow ones those that do not express *bab1*. **(F)** Distal confocal view of a mosaic L3 leg disc expressing *LAE-RFP^{ZH2A}* and harboring *bowl* mutant clones (*GFP-*). Merged Bab1 (cyan) immunostaining, RFP (red) and GFP (green) fluorescence, as well as a higher magnification of the boxed area for each marker in isolation in (F'), (F") and (F"'), respectively, are shown. Mutant clones are detected as black areas, owing to the loss of GFP. White arrows indicate pretarsal *bowl-/-* clones ectopically expressing both *bab1* and *LAE-RFP^{ZH2A}* (*bab2*).

development, sexually-dimorphic larval somatic gonad formation, salivary glue gene repression, female oogenesis, wing development as well as distal leg (tarsal) and antennal segmentation [15,17,21–28]. In addition to abdominal AE and DE, two other *bab* enhancers, termed CE and LAE (see Fig 1A), have been characterized, which recapitulate *bab2* expression in embryonic cardiac cells and developing distal leg (tarsus) as well as antennal cells, respectively [21,25,29]. However, while *bab1* and *bab2* are co-expressed in tarsal cells [15], contribution of the LAE enhancer to *bab1* regulation in the developing leg has not been yet investigated.

Adult T1-3 legs, on the pro-, meso- and meta-thoraces, respectively, are derived from distinct mono-layered epithelial cell sheets, organized as sac-like structures, called leg imaginal discs (hereafter simply referred to as leg discs) [30–32]. Upon completion of the third-instar larval stage (L3), each leg disc is already patterned along the proximo-distal (P-D) axis through regionalized expression of the Distal-less (Dll), Dachshund (Dac) and Homothorax (Hth) transcriptional regulators in the distal (center of the disc), medial and proximal (peripheral) regions, respectively [30]. The five tarsal (ts1-5) and the single pretarsal (distalmost) segments are patterned through genetic cascades mobilizing transcription factors, notably the distal selector protein Dll and the tarsal Rotund protein as well as nuclear effectors of Notch and Epidermal Growth Factor Receptor (EGFR) signaling, i.e., Bowl and C15, respectively [30,31].

Whereas both *bab* genes are required for dimorphic abdominal pigmentation traits and somatic gonad specification [17,26], only *bab2* is critical for tarsal segmentation [15]. While *bab1* loss-of-function legs are apparently wild-type, a protein null allele (*bab^{AR07}*) removing *bab2* (in addition to *bab1*) gene activity causes shortened legs owing to ts2-5 tarsal fusions as well as P-D homeotic transformations as seen by the appearance of a few up to several ectopic sex comb teeth in ts4, ts3 and ts2 segments, respectively, in males [15]. While the two *bab* genes are co-expressed within ts1-4 cells, *bab2* is expressed more proximally than *bab1* in ts1, and in a graded manner along the P-D leg axis in ts5 [15]. We previously showed that *bab2* expression in distal leg (and antennal) tissues is governed by a 567-basepair (bp) long CRE/enhancer (termed LAE for "Leg and Antennal Enhancer") which is located in between the *bab1-2* transcription units (Fig 1A) [21,29]. However, LAE enhancer contribution to *bab1* versus *bab2* regulation in the developing distal legs remains to be investigated.

Here, we show that *bab1* expression in the developing distal leg depends on the Rotund, Bowl and C15 proteins, three transcription factors known to regulate directly *bab2* expression, by binding to dedicated LAE sequences [21,29]. LAE excision by CRISPR/Cas9-mediated genome editing indicates that this enhancer is required but not sufficient for both *bab1* and *bab2* regulation and, more unexpectedly, is required also for their differential expression along the P-D leg axis. Phylogenomics analyses indicate that LAE sequences have been fixed early on during dipteran evolution, well before emergence of the *bab* complex in the Cyclorrhapha sublineage. This work illuminates how a transcriptional enhancer from tandem gene duplicates underwent evolutionary changes to diversify their respective tissue-specific gene expression pattern.

## Results

### The tandem *bab1-2* gene paralogs are co-regulated in the developing distal leg

In addition to the distal selector homeodomain (HD) protein Distal-less, we and others have previously shown that the C15 HD protein (homeoprotein) as well as Rotund and Bowl Zinc-Finger (ZF) transcription factors (TFs) bind dedicated sequences within LAE to ensure precise *bab2* expression in four concentric tarsal rings within the leg discs (Fig 1B) [21,29]. *bab1-2* are co-expressed in ts2-4 tarsal segments, while *bab2* is specifically expressed in ts5 and more

proximally than *bab1* in ts1, both in a graded manner along the P-D leg axis (Figs 1C and S1A) [15]. Given *bab1-2* co-expression in ts1-4, we first asked whether *C15*, *rotund* and *bowl* activities are also controlling *bab1* expression in the developing distal leg. To this end, we compared Bab1 expression with that of X-linked reporter genes faithfully reproducing the *bab2* expression pattern there [21,29], in homozygous mutant leg discs for a null *C15* allele or in genetically-mosaic leg discs harboring *rotund* or *bowl* loss-of-function mutant cells (Fig 1D–1F).

*C15* is specifically activated in the distalmost (center) part of the leg disc giving rise to the pretarsal (pt) segment (see Fig 1B) [33,34]. We have previously shown that the C15 homeoprotein down-regulates directly *bab2* to restrict its initially broad distal expression to the tarsal segments [29]. Bab1 expression analysis in a homozygous *C15* mutant leg disc revealed that both *bab1* and *LAE-RFP^{ZH2A}* (*bab2*) are similarly de-repressed in the pretarsus (Fig 1C and 1D).

In contrast to *C15*, *rotund* expression is restricted to the developing tarsal segments [35] and the transiently-expressed Rotund ZF protein contributes directly to *bab2* up-regulation in proximal (ts1-2) but has no functional implication in distal (ts3-5) tarsal cells [21]. Immunostaining of genetically-mosaic leg discs at the L3 stage revealed that *bab1* is cell-autonomously down-regulated in large *rotund* mutant clones in ts1-2, but not in ts3-4 segments (Fig 1E), as it is the case for *LAE-GFP^{ZH2A}* reflecting *bab2* expression. Lastly, we examined whether the Bowl ZF protein, a repressive TF active in pretarsal but not in most tarsal cells, is down-regulating *bab1* expression there [36], like *bab2* [29]. Both *bab1* and *LAE-RFP^{ZH2A}* (*bab2*) appeared cell-autonomously de-repressed in *bowl* loss-of-function pretarsal clones (Fig 1F).

In addition to loss-of-function, we also conducted gain-of-function experiments for *bowl* and *rotund*. Given Bowl TF instability when overexpressed, *bowl* gain-of-function has been achieved by down-regulating *lines* which (i) encodes a related but antagonistic ZF protein destabilizing nuclear Bowl and (ii) is specifically expressed in the tarsal territory [36]. As previously shown for *LAE-GFP^{ZH2A}* (and *bab2*) expression, nuclear Bowl stabilization in the developing tarsal region appears sufficient to down-regulate cell-autonomously *bab1* (S1C Fig). Prolonged expression of the Rotund protein in the entire distal part of the developing leg disc, i.e., tarsal in addition to pretarsal primordia, induces ectopic *bab1* expression in the presumptive pretarsal territory, as previously shown for *bab2* albeit with some differences in proximal-most GFP+ cells (S1B Fig, differentially-expressing cells are indicated with arrows), thus suggesting differential sensitivity of the two gene duplicates to Rotund TF levels (see discussion).

Taken together, these data indicate that the C15, Bowl and Rotund transcription factors, previously shown to interact physically with specific LAE sequences and thus to regulate directly *bab2* expression in the developing distal leg, are also controlling *bab1* expression there. These results suggest that the limb-specific intergenic LAE enhancer activity regulates directly both *bab* genes.

## LAE activity regulates both *bab1* and *bab2* paralogs along the proximo-distal leg axis

To test the role of LAE in regulating both *bab1* and *bab2*, we deleted precisely the LAE sequence through CRISPR/Cas9-mediated genome editing (see Materials and Methods) (Fig 2A). Two independent 3L chromosomal deletion events (termed *ΔLAE-M1* and *-M2*; see S2A Fig for deleted DNA sequences) were selected for phenotypic analysis. Both deletion mutants are homozygous viable and give rise to fertile adults with identical fully-penetrant distal leg phenotypes, namely ectopic sex-comb teeth on ts2 (normally only found on ts1) tarsal segment in the male prothoracic (T1) legs (Fig 2B), which are typical of *bab2* hypomorphic alleles [15].

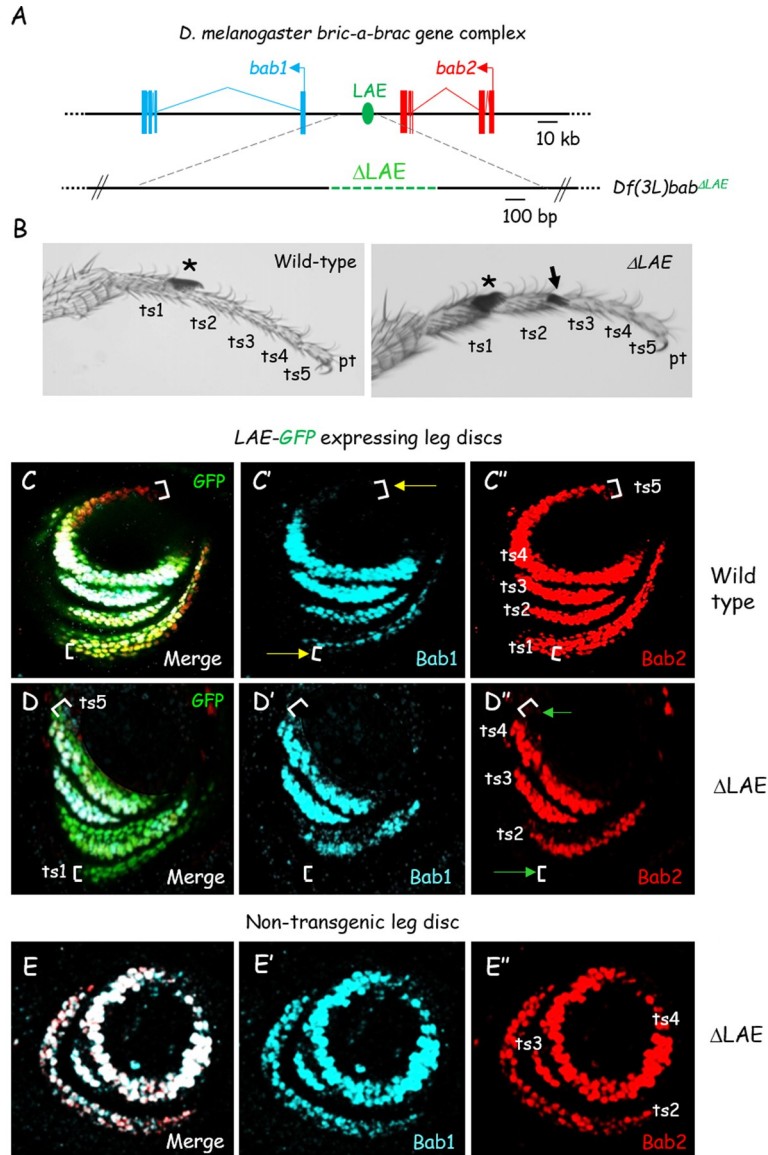

**Fig 2. LAE is differentially required for *bab1* and *bab2* expression in the developing leg. (A)** Schematic view of the *Dmel bab* locus on the 3L chromosomal arm (Chr*3L*). The tandem *bab1* and *bab2* transcription units (filled boxes and broken lines represent exons and introns, respectively) and the intergenic LAE enhancer (in green) are depicted as in Fig 1A. The small CRISPR/Cas9-mediated chromosomal deficiency (*bab^ΔLAE^*) is depicted in beneath (deleted LAE is depicted as a green broken line). **(B)** Photographs of wild-type (left) and homozygous *bab^ΔLAE^* (right) T1 distal legs from adult males. The regular sex-comb (an array of about 10 specialized bristles on the male forelegs) on distal ts1 is indicated with asterisks, while ectopic sex-comb bristles on distal ts2 from the mutant leg is indicated by an arrow. Note that the five tarsal segments remain individualized in homozygous *bab^ΔLAE^* mutant legs. **(C-D)** Confocal views of wild-type (C) and homozygous *bab^ΔLAE^* mutant (D) L3 leg discs expressing *LAE-GFP^ZH2A^*. Merged GFP fluorescence (green), Bab1 (cyan) and Bab2 (red) immunostainings, as well as the two latter in isolation in (C'-D') and (C''-D''), respectively, are shown. The respective ts1-5 fields are indicated in C''. Brackets in C-C'' show positions of GFP+ ts1 and ts5 cells expressing *bab2* in a paralog-specific manner (yellow arrows in C' indicate *bab2*-expressing GFP+ cell rows neither expressing *bab1*). Brackets in D-D'' show that neither *bab2* nor *bab1* are expressed in GFP+ ts1 and ts5 mutant cells (as indicated in D'' by green arrows). **(E)** Confocal view of a homozygous *bab^ΔLAE^* mutant L3 leg disc non-expressing the X-linked *LAE-GFP^ZH2A^* reporter. Note that Bab1-2 are strictly co-expressed in three instead of four cell rings, consistently with the pattern observed in presence of the *LAE-GFP^ZH2A^* construct.

The *ΔLAE-M1* allele was selected for detailed phenotypic analyses and is below referred to as *bab^{ΔLAE}*.

First, we quantified *bab1* and *bab2* mRNAs prepared from dissected wild-type and homozygous *bab^{ΔLAE}* mutant leg discs. As shown in S2B Fig, both mRNAs were detected in mutant discs, although *bab1* levels were two times lower than wild-type. Second, Bab1-2 expression patterns were analyzed in homozygous *bab^{ΔLAE}* leg discs. To identify leg cells that should normally express *bab2*, we used the *LAE-GFP^{ZH2A}* reporter. In homozygous *bab^{ΔLAE}* mutant leg discs, *bab2*-specific expression in proximalmost ts1 and ts5 cells (see Fig 1C) is no longer observed (Fig 2C and 2D). Furthermore, shared expression of both gene duplicates in distalmost ts1 cells is no longer detectable in *bab^{ΔLAE}* mutant discs. Nevertheless, maintenance of *bab1-2* co-expression in ts2-4 mutant cells indicates that additional *cis*-regulatory region(s) acting redundantly with the LAE enhancer must be present within the *bab* locus on the third chromosome. To exclude possible "transvection" effects of the X-linked *LAE-GFP^{ZH2A}* construct across different chromosomes [37], we also examined Bab1-2 expression patterns in homozygous *bab^{ΔLAE}* leg discs in the absence of the *LAE-GFP^{ZH2A}* reporter. As shown in Fig 2E, in the homozygous *bab^{ΔLAE}* mutant both *bab* genes are only (co-)expressed in ts2-4 cells and *bab2* remains no longer specifically expressed in ts1 an ts5 cells, ruling out a *trans*-chromosomal effect of the *LAE-GFP^{ZH2A}* transgene.

Taken together, our data indicate that intergenic LAE enhancer activity regulates both *bab* gene duplicates, being (i) required for *bab1-2* co-expression in distal ts1, (ii) dispensable for their co-expression in ts2-4, suggesting the presence of redundant *cis*-regulatory information and (iii) critically required for *bab2*-specific tarsal expression both proximally and distally (in ts1 and ts5, respectively). Thus, the LAE enhancer governs both shared and paralog-specific expression of the *bab1-2* gene duplicates.

## Chromatin features predict limb-specific *cis*-regulatory elements within *bab1*

Since LAE appeared dispensable for *bab1* and *bab2* co-expression in ts2-4 cells, our data suggested the existence of other redundant *cis*-regulatory elements. We sought to identify *cis*-regulatory information acting redundantly with LAE by taking advantage of available genome-wide chromatin features and High-throughput chromosome conformation Capture (Hi-C) experiments performed from L3 leg or eye-antennal discs (Fig 3). *bab1* and *bab2* are indeed co-expressed in distal antennal cells within the composite eye-antennal imaginal disc [15]. A topologically-associating domain covering the entire *bab* locus was detected in Hi-C data from eye-antennal discs (Fig 3A) [38], revealing particularly strong interactions between *bab1-2* promoter regions.

We then used published genome-wide data from Chromatin Immuno-Precipitation (ChIP-Seq) and Formaldehyde-Assisted Isolation of Regulatory Elements (FAIRE-Seq), as well as Assay for Transposase-Accessible Chromatin (ATAC-Seq) experiments [38–41], looking for active enhancer marks (H3K4me1 and H3K27Ac) and nucleosome-depleted chromatin regions (thus accessible to transcription factors), respectively. In the eye-antennal disc active enhancer signatures are mainly associated with a ~15-kb-long genomic region encompassing the *bab1* promoter, first exon and part of its first intron (Fig 3B). Note that LAE is also accessible to transcription factors and carries H3K4me1 marks, consistently with its enhancer activity characterized in distal antennal cells [21].

To investigate further the role of this putative enhancer region (hereafter referred to as ESR) within *bab1*, we analyzed previously-published ChIP-Seq data from L3 leg discs [42] for binding sites for Dll which is critically required to cell-autonomously activate *bab1* and *bab2*

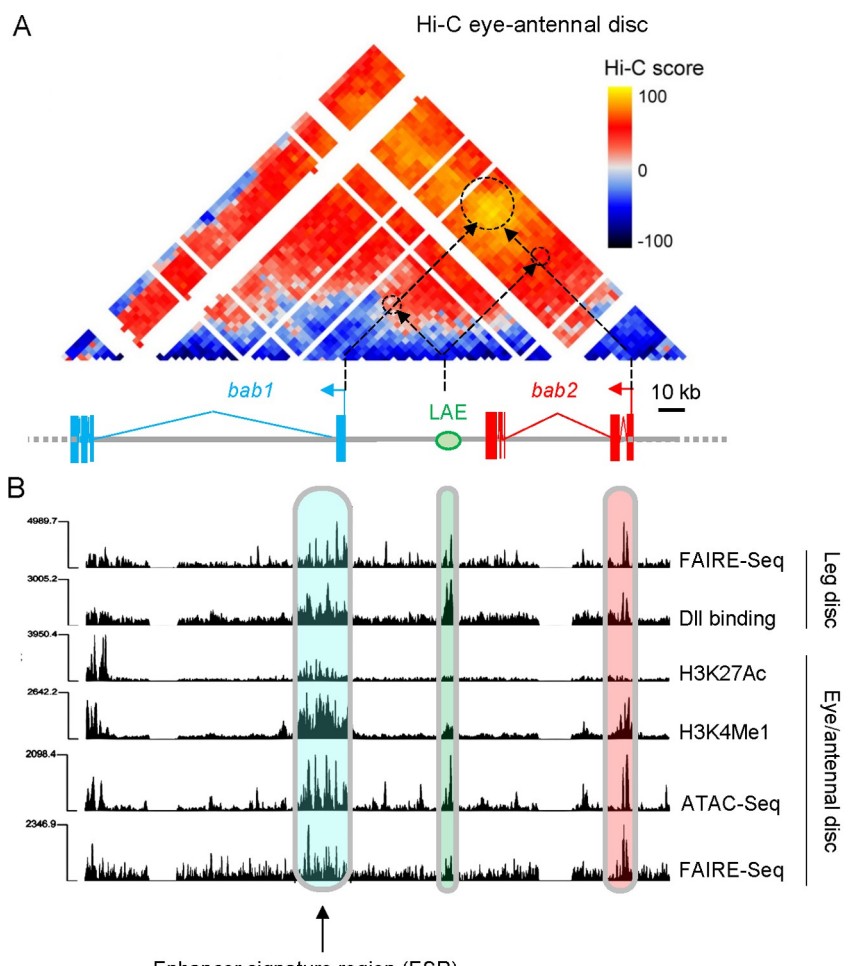

**Fig 3. Chromatin feature analyses suggest partly-redundant limb-specific regulatory information within the *bab1* transcription unit.** (**A**) Hi-C screenshot of a ~160 kb region covering the *Dmel bab* gene complex. Score scale is indicated on the right (yellow to dark blue from positive to negative). The tandem *bab1* and *bab2* transcription units as well as the intergenic LAE enhancer are depicted as in Fig 2A. (**B**) FAIRE-, ATAC- and/or ChIP-Seq profiles from L3 eye-antennal and leg discs. Normalized open chromatin, histone H3 post-translational modifications and Dll binding profiles are shown. The respective locations of the enhancer signature region (ESR), LAE and *bab2* promoter sequences are boxed in light blue, green and red, respectively.

[21,43,44]. In addition to expected binding over LAE [21] and the *bab2* promoter, strong Dll binding is also detected throughout ESR, including over the *bab1* promoter (Fig 3B).

Taken together, we concluded that the *bab1* transcription unit is predicted to include uncharacterized limb-specific regulatory information (i.e.., ESR) acting redundantly with the LAE enhancer.

## LAE functions together with *cis*-regulatory elements located within *bab1*

To validate the existence of regulatory information within the *bab1* locus, we performed phenotypic rescue experiments with Bacterial Artificial Chromosome (BAC) constructs covering each about 100 kb of genomic DNA. We have previously shown that a X-linked BAC construct, *BAC26B15^{ZH2A}*, encompassing *bab2* and the downstream intergenic sequence including LAE (see Fig 4A), is able to rescue (i) Bab2 expression in the tarsal primordium and (ii), distal leg phenotypes detected in homozygous animals for the protein null allele *bab^{AR07}*,

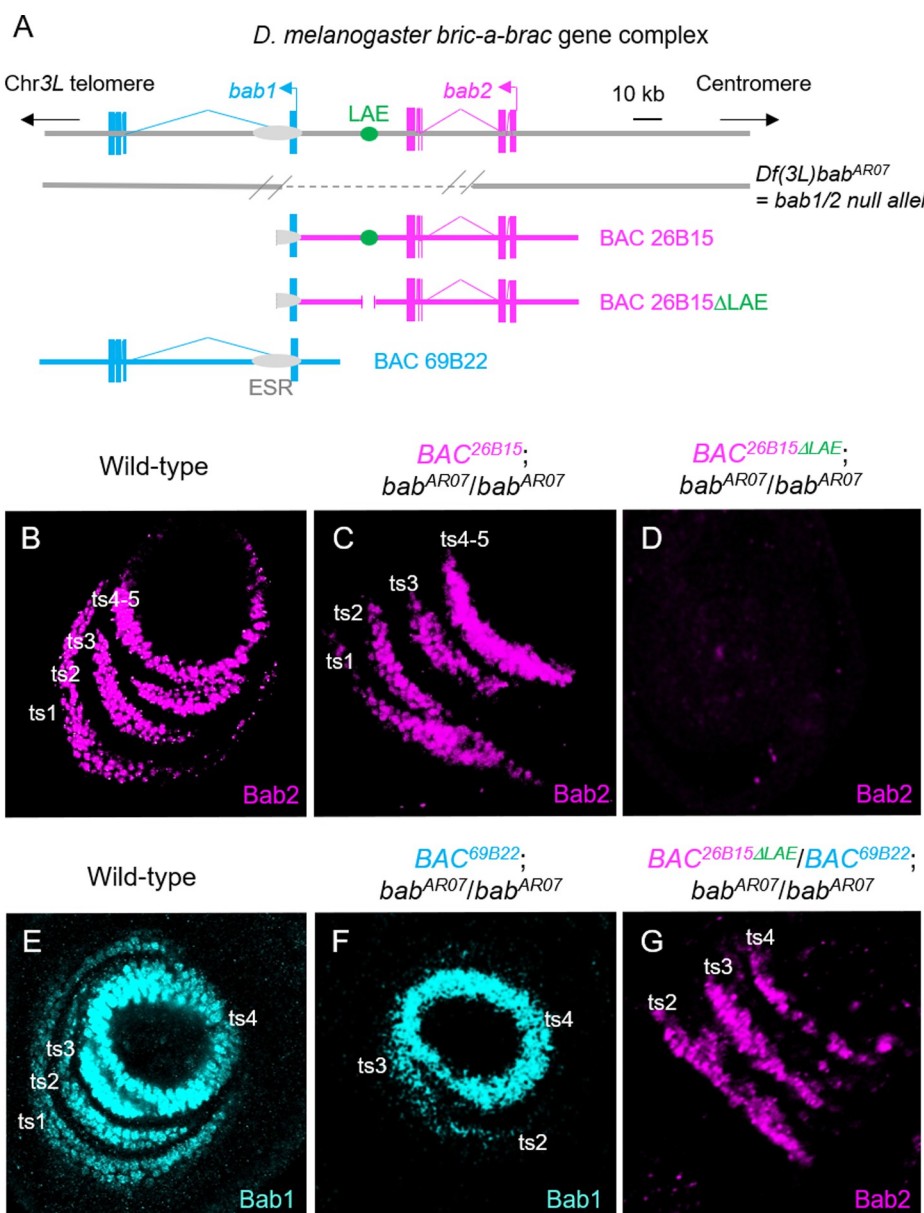

**Fig 4. The *bab1*-overlapping *BAC^69B22^* construct includes partially-redundant limb-specific *cis*-regulatory information. (A)** Chromosomal deficiency and BAC constructs covering the *bab* locus. The tandem gene paralogs and intergenic LAE are depicted as shown in Fig 1A, except that *bab2* is depicted in pink instead of red. The *bab^AR07^ 3L* chromosomal deficiency, a protein null allele removing *bab1* and *bab2* activities, is shown in beneath, with known deleted portion indicated by a dashed line. The two overlapping BAC constructs *69B22* and *26B15*, as well as a mutant derivative of the latter specifically-deleted for LAE, are shown in beneath. **(B-G)** Medial confocal views of wild-type (B, E) and homozygous *bab^AR07^* mutant (C-D and F-G) L3 leg discs, harboring singly or combined X-linked BAC construct(s) shown in (A), as indicated above each panel. Bab2 (pink) and Bab1 (cyan) immunostainings are shown. Positions of *bab1*- and *bab2*-expressing ts1-4 cells are indicated. Note stochastic *bab2* expression in (G).

affecting both Bab1 and Bab2 [15]. Conversely, a mutant *BAC26B15* construct (*BAC26-B15ΔLAE^ZH2A^*) inserted at the same genomic landing site (i.e., ZH2A on the X chromosome) and specifically lacking LAE sequence is unable to rescue Bab2 tarsal expression and leg phenotypes of *bab^AR07^* mutants (Fig 4B–4D) [21]. These data indicated that (i) in absence of

redundant *cis*-regulatory information, LAE is essential for *bab2* expression in the developing tarsus and (ii) the enhancer information redundant with LAE is located outside the genomic region covered by *BAC26B15*, which only includes the *bab1* first exon and thus lacks adjacent intronic ESR sequences.

To validate the putative regulatory information within the *bab1* transcription unit, we have tested the capacity of another BAC, *BAC69B22*, which overlaps entirely *bab1* but lacks LAE (see Fig 4A), to restore Bab1 expression in homozygous *bab^AR07* leg discs. As shown in Fig 4E and 4F, the X-linked *BAC69B22^ZH2A* construct could partially restore *bab1* expression in ts2-4 cells, indicating that it contains *cis*-regulatory information redundant with LAE activity in these tarsal segments. To test the capacity of *BAC69B22* sequences to also regulate *bab2* expression in ts2-4 cells, we placed *BAC69B22^ZH2A* across *BAC26B15ΔLAE^ZH2A*, to allow pairing-dependent *trans*-interactions (i.e., transvection; both constructs being inserted at the same ZH2A landing site) between the two X chromosomes in females. This configuration partially restored Bab2 expression in ts2-4 cells from *bab^AR07* mutant L3 leg discs, albeit in salt and pepper patterns (Fig 4G), diagnostic of transvection effects [37].

Taken together with our previous chromatin data, these genetic results are consistent with the existence within the 15 kb *bab1* ESR of uncharacterized *cis*-regulatory information capable to drive some *bab1* and *bab2* expression in distal leg tissues and acting redundantly with the LAE enhancer. The large size and complexity of this region, together with data mining from the literature, suggested that this region includes interspersed regulatory elements whose functional implication in the developing leg and antenna deserves to be studied separately.

## The *bab* complex arose from a gene duplication event in the Cyclorrhapha lineage

Both specific and common LAE enhancer activities toward *bab1* and *bab2*, as well as LAE apparent redundancy with regulatory information from the *bab1* locus provided us with a unique model to address the issue of evolutionary conservation of *cis*-regulatory landscapes governing expression of tandem paralogous genes.

To trace back the evolutionary origin of the *bab* duplication found in *D. melanogaster* (*Dmel*), we first identified proteins orthologous to *Dmel* Bab1 or Bab2, i.e., displaying an N-terminal BTB associated to a C-terminal BabCD domain (collectively referred to as BTB-BabCD proteins) [15] within highly diverse dipteran families (see Fig 5A) for which genome sequencing projects were available to us [45–47]. Two distinct BTB-BabCD proteins strongly related to *Dmel* Bab1 and Bab2, respectively, were identified in the Cyclorrhapha (higher flies) superfamily, both within the Schizophora (in Calyptratae, such as *Musca domestica* and *Glossina morsitans*, and in Acalyptratae, particularly among Drosophilidae) and Aschiza subsections (see Fig 5B). In contrast, a single BTB-BabCD protein could be identified in evolutionarily-distant dipteran species within (i) the brachyceran Empidoidea, Asiloidea and Stratiomyomorpha superfamilies (such as *Proctacanthus coquilletti*, *Condylostylus patibulatus* and *Hermetia illucens*, respectively); (ii) the Nematocera suborder families (with rare exceptions, in Psychodomorpha and Bibionomorpha, see below); (iii) other Insecta orders (e.g., Coleoptera, Hymenoptera and Lepidoptera), and in crustaceans (e.g., *Daphnia pulex*) (see S1 Data).

To analyze the phylogenetic relationships between these different Bab1/2-related proteins, their primary sequences were aligned and their degree of structural relatedness examined through a maximum likelihood analysis. As expected from an ancient duplication, cyclorrhaphan Bab1 and Bab2 paralogs cluster separately, while singleton BTB-BabCD proteins are

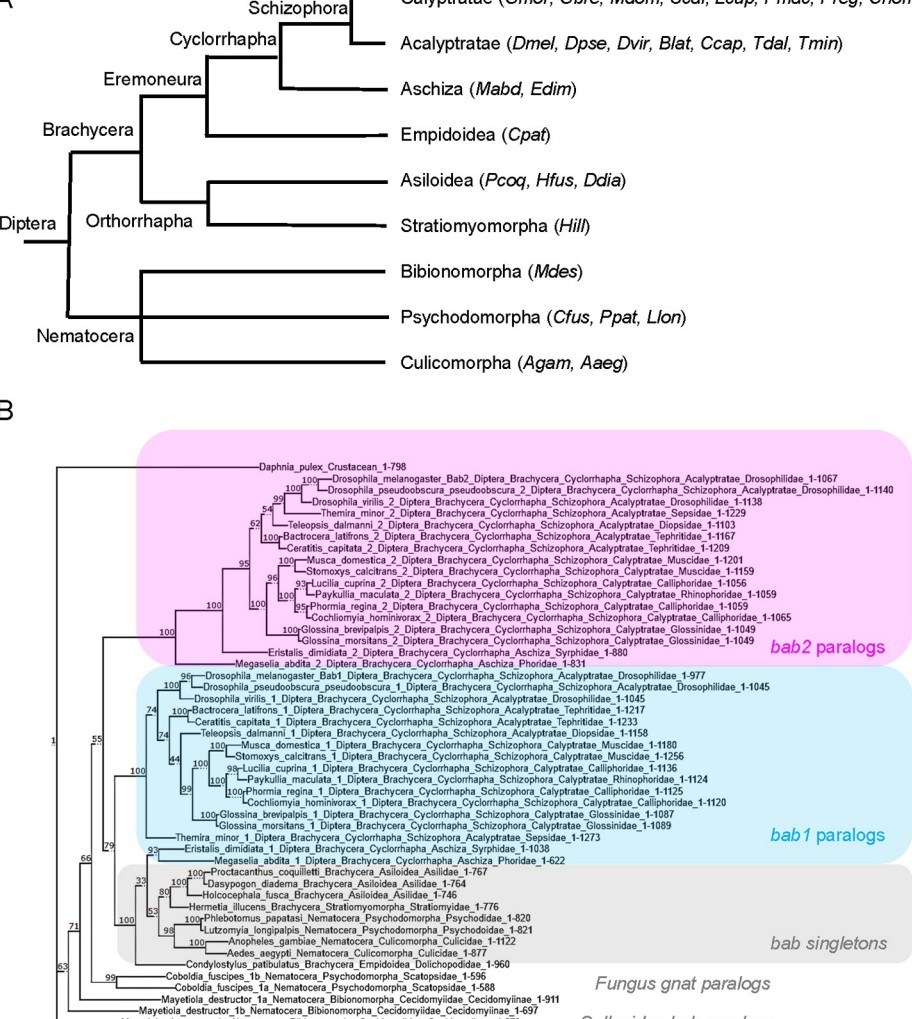

**Fig 5. Phylogenetic relationships among dipteran *bab* paralogs and orthologs. (A)** Dipteran families studied in this work and grouped according to [45]. Species abbreviations are described in S1 Data. **(B)** Phylogenetic relationships of the *bab* paralogs and orthologs inferred from a maximum likelihood consensus tree constructed from 1000 bootstrap replicates. IQ-TREE maximum-likelihood analysis was conducted under the JTT+F+R6 model. Support values (percentage of replicate trees) are shown in red. Scale bar represents substitution per site. Clustered positions of *bab1* and *bab2* paralog are shown in pink and blue, respectively, while singleton *bab* genes are depicted in grey.

more related to cyclorrhaphan Bab1 than Bab2 (Fig 5B). Branch length comparison indicates that cyclorrhaphan *bab2* paralogs have diverged more rapidly than their *bab1* twins and thus that the Bab2 clade artificially cluster separately through long-branch attraction.

Interestingly, contrary to most nematocerans, two or even three *bab* gene paralogs are present in the fungus gnat *Coboldia fuscipes* (Psychodomorpha) and the gall midge *Mayetiola destructor* (Bibionomorpha), respectively. Significantly, *M. destructor* and *C. fuscipes bab* paralogs (i) cluster separately in our phylogenetic analysis (Fig 5B) and (ii) two are arrayed in the same genomic context in both species (S3 Fig), indicating that they have likely been generated through independent gene duplication processes in the Bibionomorpha and Psychodomorpha lineages, respectively.

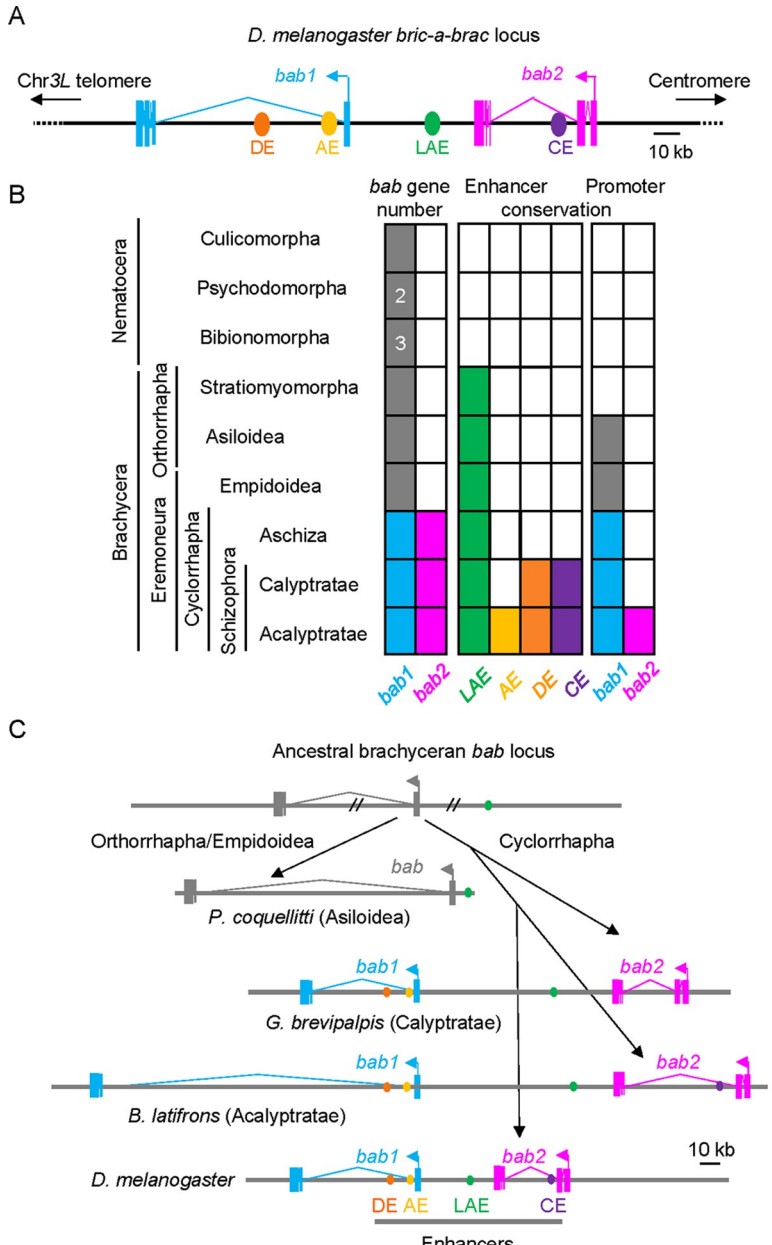

**Fig 6. Evolutionary history and enhancer sequence conservation of the *bab* locus among the Brachycera. (A)** Organization of the *Dmel bab* gene paralogs and enhancers. The locus is depicted as in Fig 1A, except that *bab2* is represented in pink instead of red. **(B)** Evolutionary conservation of the *bab* gene paralogs and enhancers among diverse dipterans. Infraorders, sections, subsections and superfamilies are indicated on the left, arranged in a phylogenetic series from the "lower" Nematocera to the "higher" Brachycera suborders. Presence of *bab* singleton or paralogs and conservation of previously-characterized enhancer sequences are indicated by filled boxes colored as depicted in (A). Presence of several *bab* paralogs in the Psychodomorpha and Bibionomorpha are indicated. **(C)** Evolutionary scenario for the *bab* locus within the Brachycera suborders. A scheme depicting chromosomal fate of an ancestral *bab* singleton gene which gives rise to derived extant orthorrhaphan *bab* singleton (Asilomorpha) and Muscomorpha-specific *bab1* and *bab2* paralogous (Calyptratae and Acalyptratae) genes. Locations of conserved enhancer sequences are shown, as depicted in (A).

Taken together, and updating a previous work [17], our phylogenomics analysis (summarized in Fig 6B and 6C) indicates that the *bab* tandem genes originated from a duplication event within the Cyclorrhapha dipteran lineage.

## LAE sequences emerged early on in the Brachycera, thus predating *bab* gene duplication

Having traced back the *bab* gene duplication raised the question of the evolutionary origin of the LAE enhancer, which regulates both *bab1* and *bab2* expression [21] (this work). We have previously shown that LAE includes three subsequences highly-conserved among twelve reference Drosophilidae genomes [48], termed CR1-3 (for Conserved Regions 1 to 3; see S4A Fig and S1 Data), of which only two, CR1 and 2, are critical for tissue-specificity [21,29]. The 68 bp CR1 includes contiguous binding sites for Dll and C15 homeoproteins, while the 41 bp CR2 comprises contiguous binding sites for Dll as well as the ZF protein Bowl (S4B and S4C Fig, respectively) [21,29].

To trace back the LAE evolutionary origin, we then systematically searched for homologous CR1-3 sequences (>50% identity) in dipteran genomes for which we identified one or two *bab* genes. Importantly, conserved LAE sequences have not been yet reported outside drosophilids. Small genomic regions with partial or extensive homologies to the CR1 (encompassing the C15 and Dll binding sites) and CR2 (particularly the Dll and Bowl binding sites) could be detected in all examined Brachycera families but not in any nematoceran (Figs 6B, S4B, and S4C). Contrary to closely-associated CR1-2 homologous sequences, no CR3-related sequence could be identified nearby, in any non-Drosophilidae species. Significantly, homologous LAE sequences are situated (i) in between the tandemly-duplicated paralogs in cyclorrhaphan species for which the entire *bab* locus sequence was available to us, suggesting an evolutionarily-conserved enhancer role, or (ii) 20 kb upstream of the *bab* singleton gene in the Asiloidea *P. coquilletti* (see Fig 6C).

Taken together, as summarized in Fig 6B and 6C, these data suggest that a LAE-like enhancer with CR1- and CR2-related elements emerged early on in the Brachycera suborder, 180–200 million years ago, and has been since fixed within or upstream the *bab* locus in the Cyclorrhapha and Asiloidea superfamilies, respectively.

## Unlike LAE, other *bab* CREs have not been conserved beyond the Cyclorrhapha

The broad LAE sequence conservation led us to also trace back the evolutionary origins of the cardiac CE, abdominal anterior AE and sexually-dimorphic DE *cis*-regulatory elements (see Fig 6A). While CE only regulates *bab2*, the AE and DE elements are predicted to govern both *bab1* and *bab2* expression in abdominal cells. Significantly, CE- and DE-related sequences could be only detected within schizophorans (excepted in Calyptratae) (Figs 6B, S5B, and S5C, respectively), whereas AE-related sequences could be readily identified within *bab* loci from drosophilids (S1 Data) but not from Aschiza, Empidoidea and Nematocera.

In conclusion, as summarized in Fig 6B and 6C, contrary to the LAE enhancer which among the Diptera emerged early on in the Brachycera suborder, other so-far identified *bab* *cis*-regulatory sequences have not been conserved beyond the Cyclorrhapha. Thus, and unlike the brachyceran LAE (CR1-2) sequences, these data indicate that other shared enhancer sequences (i.e., DE and AE) have been evolutionarily-fixed after the *bab1-2* paralog emergence.

### *bab1-2* promoter sequences have been differentially-fixed during evolution

Given the differential response of *bab1* and *bab2* to the LAE enhancer, we next analyzed the evolutionary conservation of *Dmel bab1-2* promoter core sequences (Figs 6B and S6). Both *bab* promoters are TATA-less. Whereas *bab1* has a single transcriptional initiator (Inr) element (TTCAGTC), its *bab2* paralog displays tandemly-duplicated Inr sequences (ATT-CAGTTCGT) [49,50] (S6B Fig). Both promoters display 64% sequence identity over 28 base pairs, including Inr (TTCAGT) and downstream putative Pause Button (PB; consensus CGNNCG) sequences [51] (see S6A Fig). These data suggested that (i) the duplication process having yielded *bab1-2* included the ancestral *bab* promoter and (ii) PolII pausing ability previously shown for *bab2* promoter [52–54] probably also occurs for *bab1* promoter.

Homology searches revealed that *bab1* promoter sequences have been strongly conserved in the three extant Cyclorrhapha families and even partially in some Asiloidea (e.g., *P. coquellitti*), for which a singleton *bab* gene is present (Figs 6B and S6B). In striking contrast to *bab1*, sequence conservation of the *bab2* promoter could only be detected among some Acalyptratae drosophilids (Figs 6B and S6C). In agreement with a fast-evolutionary drift for *bab2* promoter sequences, the duplicated Inr is even only detected in Drosophila group species.

Taken together, these evolutionary data (summarized in Fig 6B) indicate that, likewise for the LAE enhancer, *bab1* promoter sequences have been under strong selective pressure among the Brachycera, both in the Cyclorrhapha and Asiloidea, while paralogous *bab2* promoter sequences diverged rapidly among cyclorrhaphans. As discussed below, this evolutionary divergence may explain apparent differential activity of the LAE on each *bab* promoter.

## Discussion

In this work, we have addressed the issue of the emergence and functional diversification of enhancers from two tandem gene duplicates. Using the *Drosophila bab* locus as a model, we showed that the paralogous genes *bab1* and *bab2* originate from an ancient tandem duplication in the Cyclorrhapha lineage. The early-fixed brachyceran LAE sequence has been co-opted lately to regulate both *bab1* and *bab2* expression in a cyclorrhaphan. Furthermore, this unique enhancer is also responsible for paralog-specific *bab2* expression along the P-D leg axis. Finally, LAE governs only some aspects of *bab1-2* expression in the developing limbs because redundant *cis*-regulatory information, which remains to be characterized, is present within the *D. melanogaster bab1* gene. This work raises some hypotheses about (i) how a single enhancer can drive specificity among tandem gene duplicates, and (ii) how enhancers evolutionary adapt with distinct cognate gene promoters.

### A long-lasting enhancer sequence predating resident gene duplication

Our comprehensive phylogenomics analyses from highly diverse Diptera families indicate that the *bab* complex has been generated through tandem duplication from an ancestral singleton gene within the Cyclorrhapha (i.e., higher flies), about 100–140 years ago. This result contrasts with published data reporting that the duplication process having yielded the tandem *bab* genes occurred much earlier in the Diptera lineage leading to both the Brachycera (true flies; i.e., with short antenna) and Nematocera (long horned "flies", including mosquitos) suborders [17]. In fact, tandem duplication events implicating the *bab* locus did occur in the Bibionomorpha, as reported [17], and even in the Psychodomorpha with three *bab* gene copies (Figs 4, 5, and S3), but our phylogenetic analysis supports independent events. Thus, within the emerging dipteran lineages, the ancestral *bab* singleton gene had a high propensity to duplicate locally.

Gene duplication is a major source to generate phenotypic innovations during evolution, through diverging expression and molecular functions, and eventually from single gene copy translocation to another chromosomal site. Emergence of tissue-specific enhancers not shared between the two gene duplicates, as well as of "shadow" enhancers, have been proposed to be evolutionary sources of morphological novelties [6,55]. In this study, we have shown a strong evolutionary conservation of LAE subsequences among brachycerans, notably its CR2 element containing Dll and Bowl binding sites (S4C Fig). This conservation suggests a long-lasting enhancer function in distal limb-specific regulation of ancestral singleton *bab* genes, which has recently been co-opted in drosophilids to allow differential *bab* gene expression.

## A shared enhancer differentially regulating two tandem gene paralogs

Here, we have shown that a single enhancer, LAE, regulates two tandem gene paralogs at the same stage and in the same expression pattern. How can this work? It has been proposed that enhancers and their cognate promoters are physically associated within phase-separated nuclear foci composed of high concentrations of TFs and proteins from the basal RNA polymerase II initiation machinery inducing strong transcriptional responses [56,57]. Our Hi-C data from eye-antennal discs show a strong interaction between *bab1* and *bab2* promoter regions (Fig 3), suggesting that both *bab* promoters could be in close proximity within such phase separated droplets, thus taking advantage of shared transcriptional regulators and allowing concerted gene regulation. In contrast, no strong chromosome contacts could be detected between LAE and any of the two *bab* promoter regions, indicating that this enhancer is not stably associated to the *bab2* or *bab1* promoter in the eye-antennal disc (where only the antennal distal part expresses both genes). It would be interesting to gain Hi-C data from leg discs, in which the *bab1-2* genes are much more broadly expressed.

In addition to being required for *bab1-2* co-expression in proximal tarsal segments, we showed here that the LAE enhancer is also responsible for paralog-specific *bab2* expression along the proximo-distal leg axis. While it has been proposed that expression pattern modifications occur through enhancer emergence, our present work indicates that differential expression of two tandem gene paralogs can depend on a shared pre-existing enhancer (i.e., LAE). How this may work? Relative to its *bab1* paralog, *bab2* tarsal expression extends more proximally within the Dac-expressing ts1 cells [43] and more distally in the ts5 segment expressing nuclear Bowl protein. Furthermore, both Dac and Bowl proteins have been proposed to act as *bab2* (and presumably *bab1*) repressors [29,36,58]. CRISPR/Cas9-mediated LAE excision allowed us to establish that this enhancer is critically required for paralog-specific *bab2* expression proximally and distally, in ts1 and ts5 cells, respectively. In this context, we and others have previously proposed that transiently-expressed Rotund activating TF may antagonize Bowl (and eventually Dac) repressive activity to precisely delimit *bab2* expression among ts1 cells [21,58]. Given that *bab1-2* are distinctly expressed despite being both regulated by Bowl and Rotund, we propose that paralog-specific LAE activity depends on privileged interactions with *bab2* promoter sequences. Thus, we speculate that the *bab2* promoter responds to Rotund transcriptional activity differently from its *bab1* counterpart. Consistent with this view, ectopic Rotund expression reveals differential regulatory impacts on the two *bab* gene promoters (S1B Fig). We envision that this could occur through specific interactions between LAE-bound TFs (e.g., Rotund) and dedicated proteins within the PolII pre-initiation complex stably-associated to the *bab2* core promoter.

## Differential enhancer-promoter interplay through evolutionary changes?

Despite that sequence homologies between both promoters (consistent with an ancient duplication event mobilizing the ancestral singleton *bab* promoter) are still detectable, it is

significant that the *bab2* promoter evolves much faster than its *bab1* counterpart. While the *bab1* promoter sequence has been strongly conserved among cyclorrhaphans, with sequence homologies with brachyceran singleton *bab* promoters, the *bab2* promoter sequence has only been fixed recently among Drosophilidae, notably through the Initiator (Inr) sequence duplication, indicating very fast evolutionary drift after the gene duplication process which yielded the *bab1/2* paralogs. We envision that this evolutionary ability has largely contributed to allow novel expression patterns for *bab2*, presumably through differential enhancer-promoter pairwise interplay.

## Materials and methods

### Fly stocks, culture and genetic manipulations

*D. melanogaster* stocks were grown on standard yeast extract-sucrose medium. The *vasa*-PhiC31 ZH2A *attP* stock (kindly provided by F. Karch) was used to generate the *LAEpHsp70-GFP* reporter lines and the *BAC69B22* construct as previously described [21]. *LAE-GFP* and *LAE-RFP* constructs inserted on the ZH2A (X chromosome) or ZH86Fb (third chromosome) *attP* landing platforms, and displaying identical expression patterns, have been previously described [21,29]. $C15^2$/TM6B, $Tb^1$ stock was kindly obtained from G. Campbell. Mutant mitotic clones for null alleles of *bowl* and *rotund* were generated with the following genotypes: *y w LAE-GFP*; $DllGal4^{EM2012}$, *UAS-Flp/+; FRT82B, Ub-RFP/FRT82B $rn^{12}$* (i.e., *rn* mutant clones are RFP negative; Fig 1E) and *y w LAE-RFP*; $DllGal4^{EM2012}$, *UAS-Flp/+; Ub-GFP, FRT40A/$bowl^1$ FRT40A* (i.e., *bowl* mutant clones are GFP negative; Fig 1F), respectively. Rotund protein gain-of-function within the *Dll*-expressing domain was obtained with the following genotype: *y w LAE-GFP*; $DllGal4^{EM2012}$; $UAS-Rn^1$/+. The $Dll^{EM212}$-Gal4 line was provided by M. Suzanne, while the $UAS-Rn^1$ line was obtained from the Bloomington stock center. "Flip-out" (FO) mitotic clones over-expressing dsRNA against *lines* were generated by 40 mn heat shocks at 38°C, in mid-late L2 to early-mid L3 larvae of genotypes: *y w LAE-RFP hsFlp*; *UAS-dsRNAlines/pAct>y+>Gal4, UAS-GFP* (i.e., FO clones express GFP in S1C Fig). The *UAS-dsRNA* stock used to obtain interfering RNA against *lines* (#40939) was obtained from the Bloomington stock Center.

### Immuno-histochemistry and microscopy

Leg discs were dissected from wandering (late third instar stage) larvae (L3). Indirect immuno-fluorescence was carried out as previously described [21] using a LEICA TCS SP5 or SPE confocal microscope. Rat anti-Bab2 [15], rabbit anti-Bab1 [18], rabbit ant-Dll [59], rabbit anti-Bowl [58], and rabbit anti-C15 [34] antibodies were used at 1/2000, 1/500, 1/200, 1/1000 and 1/200, respectively.

### CRISPR/Cas9-mediated chromosomal deletion

Guide RNAs (gRNAs) were designed with CHOPCHOP at the Harvard University website (https://chopchop.cbu.uib.no/). Four gRNA couples were selected that cover two distinct upstream and downstream LAE positions: TGCGTGGAGCCTTCTTCGCCAGG or TGGAGCCTTCTTCGCCAGGCCGG; and TATACTGTTGAGATCCCATGCGG or TTAGGCGCACATAAGGAGGCAGG (the PAM protospacer adjacent motif sequences are underlined), respectively. Targeting tandem chimeric RNAs were produced from annealed oligonucleotides inserted into the pCFD4 plasmid, as described in (http://www.crisprflydesign.org/). Each pCFD4-LAE-KO construct was injected into 50 *Vasa-Cas9* embryos (of note the *vasa* promoter sequence is weakly expressed in somatic cells). F0 fertile adults and their F1

progeny, with possible somatic LAE-deletion events and candidate mutant chromosomes (balanced with *TM6B*, *Tb*), respectively, were tested by polymerase chain reactions (PCR) with the following oligonucleotides: AGTTTTTCATCCCCCTTCCA and GTATTTCTTTGCCTTGCC ATCG (predicted wild-type amplified DNA: 2167 base pairs).

## Quantitative RT-PCR analysis

T1-3 leg imaginal discs were dissected from homozygous *white^1118^* and *bab^ΔLAE-M1^* late L3 larvae in PBS 0.1% Tween. 50 discs of each genotype were collected and frozen in nitrogen. Total messenger RNAs were purified using RNeasy kit (Qiagen) and reverse transcribed (RT) by SuperScript II (ThermoFisher) and quantified by quantitative PCR (qPCR) using the ΔΔCt method from Bio-Rad CFX Manager 3.1 software [60]. *bab1*, *bab2*, *Rpl32*, *Gpdh1* or *Mlc-c* cDNA levels were monitored by qPCR using the following oligonucleotides: Bab1Fw: CGCCCAAGAGTAACAGAAGC; Bab1Rev: TCTCCTTGTCCTCGTCCTTG; Bab2Fw: CTGCAGGATCCAAGTGAGGT; Bab2Rev: GACTTCACCAGCTCCGTTTC; Rpl32Fw: GACGCTTCAAGGGACAGTATCTG; Rpl32Rev: AAACGCGGTTCTGCATGAG; Gpdh1 Fw: TCTTCCAGGCGAACCACTTC; Gpdh1Rev: AGGCCACGATGTTCTTGAGG; Mlc-cFw: GCGGTTATATCTCCTCCGCC; Mlc-cRev: CGTAGTTGATGTTGCCCTGCA: Wilcoxon test was performed to evaluate the difference between samples.

## Homology searches, sequence alignments and phylogenetic analyses

Homology searches were done at the NCBI Blast site (https://blast.ncbi.nlm.nih.gov/Blast.cgi). Protein or nucleotide sequence alignments were done using MAFFT (Multiple Alignment using Fast Fourier Transform) (https://mafft.cbrc.jp/alignment/server/). Phylogenetic relationships were inferred through a maximum likelihood analysis with W-IQ-Tree (http://iqtree.cibiv.univie.ac.at/), using JTT+F+R6 as a substitution model, and visualized with the ETE toolkit (http://etetoolkit.org/treeview/).

## Transcription factor binding prediction

DNA binding predictions were done using the motif-based sequence analysis tool TomTom from the MEME suite (https://meme-suite.org/meme/tools/tomtom) and the Fly Factor Survey database (http://mccb.umassmed.edu/ffs/).

## Supporting information

**S1 Fig. Contrary to Bowl, Rotund gain-of-function differentially affects *bab1* and *LAE-GFP* (*bab2*) expression.** (A) The *bab1* paralog is expressed in a subset of *LAE-GFP^ZH2A^* (*bab2*)-expressing cells, both proximally and distally within the developing tarsus. Merged Bab1 (red) immunostaining and GFP fluorescence (green) as well as each marker in isolation in (A') and (A"), respectively, are shown for a wild-type L3 leg disc expressing *LAE-GFP^ZH2A^* (medial confocal view). Positions of *LAE-GFP^ZH2A^* (*bab2*)-expressing ts1-5 cells and of the non-expressing pre-tarsal (pt) cells are indicated in (A) and (A'). Brackets indicate paralog-specific expression in *bab2*-expressing (GFP+) ts1 and ts5 cells, as detected as green- instead of yellow-colored cells in (A) (see also white arrows in (A')). Of note, *bab1* is only expressed in distal ts1 cells, while *LAE-GFP^ZH2A^* (*bab2*) expression extends proximally. (B) Rotund TF gain-of-function within the developing *Dll*-expressing cells differentially activates the *bab* gene paralogs along the P-D leg axis. Merged Bab1 (red) immunostaining and GFP (green) fluorescence, as well as each marker in isolation in (B') and (B"), respectively, are shown for a leg disc dissected from a L3 larvae harboring both *UAS-Rn* and *Dll^EM212^-Gal4* transgenes. Contrary to

a distal domain (circled with a dashed line) in which both *bab1* and *LAE-GFP$^{ZH2A}$* (*bab2*) are strictly co-expressed, many proximalmost *Dll*-expressing GFP+ cells neither activate *bab1* (some are indicated by white arrows). **(C)** Ectopic Bowl TF stabilization, through clonal Lines protein depletion, is sufficient to down-regulate both *bab1* and *LAE-GFP$^{ZH2A}$* (*bab2*) expression. Merged Bab1 (cyan) immunostaining, RFP (red) and GFP (green) fluorescence, as well as the two former markers in isolation in (C') and (C"), respectively, are shown for a L3 leg disc expressing *LAE-RFP$^{ZH2A}$*. Flip-out (FO) mitotic clones are detected through GFP expression in (C), and are circled with dashed lines in (C') and (C"). Within the developing tarsus Bowl stabilization leads to cell-autonomous repression of both *bab1* and *LAE-RFP$^{ZH2A}$* (*bab2*). (TIF)

**S2 Fig. LAE deletion mutant behaves as a hypomorphic allele. (A)** Targeted deletion of the LAE with CRIPSR/Cas9 genome editing. The sequences flanking LAE from the wild-type (Wt) and six deleted chromosomes (M1-6) are shown. LAE sequences are depicted in orange while exogenous sequences in mutant chromosomes are indicated by distinctly-colored lower case letters (unmodified nucleotides are upper case ones). **(B)** Overall *bab1-2* expression from wild-type and homozygous *bab$^{ΔLAE}$* L3 leg discs, as determined from reverse transcription quantitative PCR analyses. mRNA levels are normalized from expression of three housekeeping genes: *Rpl32*, *Mlc-c* and *Gpdh1*. Results show the mean and the standard error of the mean of 4 independent experiments (Wilcoxon test p value < 0.05 is indicated by $^*$). (TIF)

**S3 Fig. Predicted structural organizations of *bab*-related gene complexes among nematocerans.** *bric-a-brac* paralogs from the fungus gnat *Coboldia fuscipes* (Psychodomorpha) and the gall midge *Mayetiola destructor* (Bibionomorpha), are shown. GenBank identifiers of the corresponding genomic sequences are indicated. (TIF)

**S4 Fig. LAE sequence conservation among the Brachycera. (A)** Structural conservation of the *Dmel* LAE enhancer among Drosophilidae. The locations of CR1-3 sequences, conserved among 12 reference drosophilid genomes, are boxed in green. **(B-C)** Alignments of brachyceran CR1 (B) and CR2 (C) sequences are shown. The four-letter species abbreviations are listed in S1 Data. Strictly conserved positions are indicated by white characters on a red background while partially ones conserved (>50%) are in black characters on a yellow background. The sequence LOGOs for the evolutionarily-conserved C15, Dll and Bowl binding sites are indicated above the aligned sequences. (TIF)

**S5 Fig. Cardiac and abdominal enhancer sequence conservation among schizophorans. (A)** Schematic view of the DE and CE enhancers within the *Dmel bab* locus. The tandem *bab1* (blue) and *bab2* (magenta) transcription units are depicted as in Fig 4A. Positions of the evolutionarily-conserved cores within the cardiac CE and abdominal DE sequences are shown in beneath. **(B-C)** Evolutionary conservation of CE (B) and DE (C) core sequences among schizophorans. The four-letter species abbreviations are listed in S1 Data. Strictly conserved positions are indicated by white characters on a red background while partially conserved ones (>50%) are in black characters on a yellow background. The sequence LOGOs for *bona fide* (Dsx and Abd-B) or predicted (Twist-Da, Lbe and Pan) transcription factor binding sites are shown above or below the alignments. (TIF)

**S6 Fig. *bab1-2* promoter sequence conservation among brachycerans. (A)** Sequence homology between the *Dmel* twin *bab* gene promoters. Positions of initiator (Inr) and pause button (PB) sequences are indicated above the aligned sequences. Transcription start site (TSS) is indicated by a vertical arrow. **(B-C)** Evolutionary conservation of *bab1* (B) and *bab2* (C) promoter sequences, among selected dipteran lineages (as indicated on the left side). The four-letter species abbreviations are listed in S1 Data. Strictly conserved positions are indicated by white characters on a red background while partially conserved ones are in black characters on a yellow background. Inr, PB and TSS locations are depicted as in (A).
(TIF)

**S1 Data. p.2. Abbreviations of investigated species. p.3-20. Predicted sequences for BTB-BabCD proteins**. **p.21-22. Bab1 sequence conservation among cyclorrhaphans**. **p.23-24. Bab2 sequence conservation among cyclorrhaphans**. **p.25-29 Sequence conservation between Bab1/2 paralogs** Sequence conservation between paralogous Bab1/2 proteins among cyclorrhaphans. The four-letter species abbreviations are as listed above (p.2). Strictly conserved amino-acid residues are indicated by white characters on a red background while partially conserved ones are in black characters on a yellow background. Locations of the strongly-conserved BTB and BabCD domains are indicated along the right side (see black lines). **p.30-39 Enhancer sequence conservation among Drosophilidae**. Conservation among twelve reference drosophilids of *D. melanogaster* LAE, CE, AE and DE sequences. The four-letter Drosophilidae species abbreviations are as listed below (page 2). Sequence LOGOs of (predicted) binding sites for the Dll, Bowl, C15, Rn, Pan, Lbe, Twist, Abd-B and Dsx transcription factors are depicted above or below the alignments.
(PDF)

## Acknowledgments

We thank F. Karch, M. Suzanne, T.M. Williams, G. Boekhoff-Falk, S. Bray, G. Campbell, T. Kojima, F. Laski, J.L. Couderc and the Bloomington Stock Center for fly stocks and reagents. We are grateful to Alain Vincent for his proofreading of the manuscript. We thank Julien Favier for technical assistance, particularly in managing the transgenic facility. Lastly, we acknowledge Brice Ronsin and the Toulouse RIO Imaging platform.

## Author Contributions

**Conceptualization:** Henri-Marc G. Bourbon, Frédéric Bantignies, Giacomo Cavalli, Muriel Boube.

**Formal analysis:** Henri-Marc G. Bourbon, Vincent Loubiere.

**Funding acquisition:** Henri-Marc G. Bourbon, Giacomo Cavalli, Muriel Boube.

**Investigation:** Henri-Marc G. Bourbon, Mikhail H. Benetah, Emmanuelle Guillou, Luis Humberto Mojica-Vazquez, Aissette Baanannou, Sandra Bernat-Fabre, Vincent Loubiere, Muriel Boube.

**Methodology:** Henri-Marc G. Bourbon.

**Project administration:** Henri-Marc G. Bourbon, Frédéric Bantignies, Giacomo Cavalli, Muriel Boube.

**Resources:** Sandra Bernat-Fabre.

**Supervision:** Henri-Marc G. Bourbon, Frédéric Bantignies, Giacomo Cavalli, Muriel Boube.

**Validation:** Henri-Marc G. Bourbon, Mikhail H. Benetah, Emmanuelle Guillou, Luis Humberto Mojica-Vazquez, Aissette Baanannou, Sandra Bernat-Fabre, Vincent Loubiere, Frédéric Bantignies, Muriel Boube.

**Visualization:** Henri-Marc G. Bourbon, Mikhail H. Benetah, Luis Humberto Mojica-Vazquez, Aissette Baanannou, Muriel Boube.

**Writing – original draft:** Henri-Marc G. Bourbon, Muriel Boube.

**Writing – review & editing:** Henri-Marc G. Bourbon, Vincent Loubiere, Frédéric Bantignies, Muriel Boube.

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
