## [Decision Letter · Decision Letter 0]

13 Dec 2021

Dear Dr BOUBE,

Thank you very much for submitting your Research Article entitled 'A shared ancient enhancer element differentially regulates the bric-a-brac tandem gene duplicates in the developing Drosophila leg' to PLOS Genetics.

The manuscript was fully evaluated at the editorial level and by independent peer reviewers. The reviewers appreciated the attention to an important topic but identified some concerns that we ask you address in a revised manuscript. We therefore ask you to modify the manuscript according to the review recommendations. Your revisions should address the specific points made by each reviewer.

[LINK]

Yours sincerely,

Artyom Kopp

Associate Editor

PLOS Genetics

Gregory P. Copenhaver

Editor-in-Chief

PLOS Genetics

Reviewer's Responses to Questions

**Comments to the Authors:**

Reviewer #1: This revised and improved manuscript by Bourbon et al. has addressed the majority of reviewers’ suggestions and concerns. In particular, the manuscript has been streamlined to include the most convincing observations of the original manuscript, which support the authors’ conclusions that: 1) bab1/2 are co-regulated by a common set of transcription factors acting through a leg-antennal enhancer (LAE); 2) the LAE is required for a subset of both bab1/2 expression along P/D axis of legs 3); the LAE is essential for bab2-specific expression in the ts1 and ts5 leg segments 4) the LAE function is partially redundant with sequences in the 15kb ESR, and 5) bab1 and bab2 arose from a tandem duplication in the Muscomorpha lineage.

While the revised manuscript is substantially improved, most revisions involved removal of data, such as the extensive, but incomplete, analysis of the ESR (previously called the BER). This has strengthened the document and has made it easier to follow, but it might have been further improved by additional analysis, suggested by another reviewer, that tests their idea that privileged interactions between the LAE and the bab2 promoter underlies LAE-mediated bab2-specific expression. Nonetheless, their conclusions overall are well supported by substantial and compelling observations, and this work provides important new insights into the origin and regulation of duplicated genes.

Reviewer #2: The authors investigate the role and the evolution of regulatory sequences driving bab1 and bab2 expression in Drosophila leg imaginal disc in parallel with bab duplication history in Dipterans.

The authors show that the transcription factors regulating bab2 expression in leg disc via the LAE enhancer also regulate bab1 expression in leg disc suggesting that the LAE enhancer regulate both genes.

The deletion of the LAE enhancer by CRISPR/CAs9 leads to an ectopic sexcomb on the second tarsal segment, a phenotype similar to that of bab2 hypomorphic mutants. Thus it is distinct from the stronger phenotype observed in bab1 and bab2 double mutants. Only bab1 expression is reduced in leg discs of the LAE deleted mutant as measured by RT-qPCR. However expression reduction of both genes along the proximodistal axis is revealed by immunostaining. The residual and significant expression must therefore be controlled by other redundant regulatory sequences. The authors very nicely exclude a potential transvection effect caused by the LAE-GFP transgene as the residual expression is observed in the absence of this transgene.

To identify the redundant regulatory sequences driving bab1 and bab2 expression in leg discs, the authors turn to chromatin analysis. They used available HI-C data from L3 leg disc and eye antennal disc and identify a topologically associated domain (TAD) covering bab1 and interacting with bab1-2 promoters. Using FAIRE-seq, ATAC-seq and active histone marks characterizing enhancers they identify enhancer signature along a 15Kb region covering bab1. In addition, ChIP seq data identify binding sites for DLL, an activator of bab genes in leg discs, in bab1 region. The authors use successfully a transvection assay to show that the redundant regulatory sequences present in bab1 are able to activate bab2 promoter (even if bab2 expression is noisy). Thus bab1 transcription unit likely contains enhancer sequences acting redundantly with LAE in controlling the expression of bab genes in leg disc.

Rescue experiments with BAC constructs containing regions of bab2 with intact or deleted LAE in a bab1-2 double mutant indicate that in the absence of the redundant regulatory sequences, the LAE is essential for bab2 expression in leg disc. Rescue experiment of the bab1-2 double mutant with a BAC containing entirely bab1 but not LAE indicate that this BAC contains the redundant regulatory sequences controlling bab genes expression in leg disc. Because of the large size and the complexity of this region (which contains also two distinct enhancers involved in bab1-2 expression in abdominal epidermis), the authors do not investigate further this region in this article.

The authors use then an evolutionary approach to reconstruct the evolution of bab gene family and their regulatory sequences in dipterans. bab1 and bab2 paralogues were identified in muscomorpha but a single bab genes was found in more distant dipterans (with a few exceptions). A maximum likelihood phylogeny with the protein sequences of bab genes was performed. bab1 and bab2 clades do not group together. The authors interpret it as consistent with an ancient duplication of bab1-2 genes within the muscomorpha lineage (I have comments on the interpretation of the phylogeny see bellow). Independent bab duplications have occurred in biblionomorpha and psychodomorpha lineages.

Sub-regions of the LAE enhancer can be identified in all Brachycera family. It is located as in Drosophila melanogaster between bab1 and bab2 in muscomorphan species, and 20 kb upstream of the singleton bab in the asilomorphan P. coquilletti. Thus the LAE element arose early in the Brachycera suborder. Other enhancers known to regulate bab1-2 expression in other tissue cannot be traced back in species as distant from D. melanogster as for the LAE element and were likely acquired after the bab1/2 duplication.

bab1 and bab2 promoter share some conservation which suggests that the promoter of the ancestral gene was also duplicated. bab1 promoter is more conserved in dipterans than bab2 promoter, which is consistent with a more rapid evolution of bab2.

In the discussion, the authors summarize their results on the early birth of the LAE enhancer that predates bab1/2 duplication and propose how their finding and previously published data could explain the differences of bab1 and bab2 expression in leg disc, based on different interactions between the LAE enhancer and bab2 promoter.

I enjoyed reading this very interesting manuscript, clearly written and well referenced. The methodology is rigorous. Many data have been generated using different approaches, which makes this work original. It allows the authors to illustrate how regulatory sequences evolve in parallel with gene duplication. More data on the redundant regulatory sequences present in bab1 would be a nice addition to this work, but I agree that the detailed investigation of these regulatory sequences is out of the scope of this paper as the authors write, because of the size and the complexity of this region. This work will appeal to a broad readership, in particular researchers in genetics and genomics, and fully deserves publication in PLOS Genetics after minor revisions (see bellow).

Comments:

Introduction : lines 70-73, it is written that phenotypic novelties are linked to the emergence or the modification of cis-regulatory elements. This is true generally for morphological evolution, but the modification of protein sequence has been shown to play a very important role in physiological evolution. We know many examples of adaptive evolution linked to protein sequence evolution (opsin, globin, venom proteins, temperature receptors, protein involved in resistance to plant toxin, etc...). I think this distinction between morphological and physiological evolution should be made.

There are other examples in drosophila of gene complexes in which particular paralogues have been shown to be regulated by shared enhancers (achaete-scute complex, spalt complex, Enhancer of split etc...). They could be briefly presented in the introduction.

Line 83, it is written that yellow expression takes place in histoblast nests. However at the stage when yellow is expressed, the histoblast nests have finished proliferating. Left and right sides have fused. I think it is more appropriate to use the term “epidermis”.

In the introduction, the effect of bab2 loss of function on leg development could be described. Only the effect of bab1 loss of function or loss of function of both bab1 and bab2 is described.

Only one control gene was used to normalize bab1 and bab2 expression in RT-qPCR (Figure S2). It is usually advised to use several ones. I think at least one other control gene should be measured.

In Figure 3, the scale bar of 10Kb is missing.

In Figure 5A, I think it would be better to provide a tree showing evolutionary relationships of dipterans extracted from published dipteran phylogenies (such as Wiegmann et al., 2011; There are also more recent phylogenies on some Dipteran sub-groups).

The phylogenetic reconstruction using bab gene protein sequences is not enough detailed in the Material and Methods. Were highly diverging regions excluded from the analysis or were all sites used? If all sites were used, how did the authors deal with regions impossible to align?

In the phylogeny presented in Fig 5B bab1 and bab2 clades do not group together. The authors interpret it as consistent with an old duplication and a faster evolution of bab2 genes. However, if the phylogeny is taken for granted, the grouping of bab1 with bab singleton clade indicate that they are orthologous and that bab2 was lost in an ancestor of species containing only one bab gene. Thus it does not corresponds to the authors’ interpretation. However, the support for bab clade grouping with the singleton bab clade is low. Thus the more basal position of bab2 is more likely artificial and caused by long branch attraction if bab2 is evolving rapidly. I therefore agree with the authors’ conclusion but I think that this should be discussed. There is also a phylogeny of bab genes in Figure S3A. Many more species are included than in the tree from figure 5B. In this tree, bab1 and bab2 paralogues group together (although with a low support) and two bab1 paralogues group artificialy before bab1/bab2 split. This difference between the two trees should be commented.

I think that he leg phenotype with the ectopic sexcomb of the LAE CRISPR/Cas9 deleted flies (Figure S2) is very nice and should be included in Figure 2 in order to be present in the main article.

Thanks for the opportunity to review this very nice work!

Reviewer #3: The manuscript by Bourbon et al. is a revised and concise version of a manuscript I reviewed last spring. In this new submission, the authors omitted the parts of the manuscript that were ambiguous and focused on the solid parts of the study. Namely, they focus on the question of how regulatory information is rewired after gene duplication. The authors use the tandemly duplicated Drosophila bric-a-brac (bab) paralogs, bab1 and bab2 to address this question. Building on their previous work on the bab2 LAE enhancer, that drives expression in eye-antenna and leg imaginal discs, they show that LAE also regulates bab1 expression in these tissues by using the same set of transcription factors. Interestingly, they find that the LAE enhancer can drive both redundant and unique expression patterns and suggest that the unique expression patterns result from specific interactions with the promoter of each paralog. The authors performed a comprehensive phylogenomic analysis to trace the evolutionary origin of the bab genes and their regulatory sequences. They show that the LAE enhancer region predated the gene duplication event, and suggest that after the duplication, paralogous-specific regulatory connections were generated between the duplicated gene and the enhancer. This work provides interesting new insights into the mechanisms by which new regulatory linkages are form during gene duplication and I think is appropriate for publication in Plos Genetics it its current format (with two minor comments).

Minor comments:

1. Please follow the journal Figure File Requirements (For example: text within the Figures should be Ariel or Times).

2. Line 383: it would be more accurate to state that “This work raises some hypotheses” instead of “This work brings some clues…” as no functional data was presented to support these “clues”.

**Have all data underlying the figures and results presented in the manuscript been provided?**

Reviewer #1: Yes

Reviewer #2: Yes

Reviewer #3: Yes

PLOS authors have the option to publish the peer review history of their article (what does this mean?). If published, this will include your full peer review and any attached files.

Reviewer #1: No

Reviewer #2: No

Reviewer #3: No

---

## [Editor Report · Decision Letter 1]

7 Feb 2022

Dear Dr BOUBE,

We are pleased to inform you that your manuscript entitled "A shared ancient enhancer element differentially regulates the bric-a-brac tandem gene duplicates in the developing Drosophila leg" has been editorially accepted for publication in PLOS Genetics. Congratulations!

Yours sincerely,

Artyom Kopp

Associate Editor

PLOS Genetics

Gregory P. Copenhaver

Editor-in-Chief

PLOS Genetics

Comments from the reviewers (if applicable):

**Data Deposition**

http://datadryad.org/submit?journalID=pgenetics&manu=PGENETICS-D-21-01427R1

**Press Queries**

---

## [Editor Report · Acceptance letter]

10 Mar 2022

PGENETICS-D-21-01427R1 

A shared ancient enhancer element differentially regulates the bric-a-brac tandem gene duplicates in the developing Drosophila leg 

Dear Dr BOUBE, 

We are pleased to inform you that your manuscript entitled "A shared ancient enhancer element differentially regulates the bric-a-brac tandem gene duplicates in the developing Drosophila leg" has been formally accepted for publication in PLOS Genetics! Your manuscript is now with our production department and you will be notified of the publication date in due course.

With kind regards,

Katalin Szabo

PLOS Genetics

On behalf of:
